# The Role of Complement Dysregulation in Glaucoma

**DOI:** 10.3390/ijms25042307

**Published:** 2024-02-15

**Authors:** Cindy Hoppe, Meredith Gregory-Ksander

**Affiliations:** 1Schepens Eye Research Institute of Mass Eye and Ear, Department of Ophthalmology, Harvard Medical School, Boston, MA 02114, USA; cindy_hoppe@meei.harvard.edu; 2Animal Physiology/Neurobiology, Department of Biology, Friedrich-Alexander-Universität Erlangen-Nürnberg, 91058 Erlangen, Germany

**Keywords:** glaucoma, complement system, inflammation

## Abstract

Glaucoma is a progressive neurodegenerative disease characterized by damage to the optic nerve that results in irreversible vision loss. While the exact pathology of glaucoma is not well understood, emerging evidence suggests that dysregulation of the complement system, a key component of innate immunity, plays a crucial role. In glaucoma, dysregulation of the complement cascade and impaired regulation of complement factors contribute to chronic inflammation and neurodegeneration. Complement components such as C1Q, C3, and the membrane attack complex have been implicated in glaucomatous neuroinflammation and retinal ganglion cell death. This review will provide a summary of human and experimental studies that document the dysregulation of the complement system observed in glaucoma patients and animal models of glaucoma driving chronic inflammation and neurodegeneration. Understanding how complement-mediated damage contributes to glaucoma will provide opportunities for new therapies.

## 1. Introduction

Glaucoma is a neurodegenerative disease, characterized by the death of retinal ganglion cells (RGCs) and degeneration of their axons. Over 76 million people worldwide suffer from glaucoma, and this number is expected to increase to approximately 111.8 million by 2040 [1].

The primary risk factor for glaucoma is high intraocular pressure (IOP), and the current treatment approach focuses on lowering IOP. However, lowering IOP alone only slows disease progression, but it does not completely stop progression, leading to visual field loss in many patients [2,3]. Primary open-angle glaucoma is the most common form of glaucoma. Interestingly, some individuals may develop high IOP but do not go on to develop glaucoma, while others with normal IOP can develop glaucoma, known as normal-tension glaucoma, which accounts for 30–40% of all cases [4].

The pathology of glaucoma is not well understood. While elevated IOP is the main risk factor, other factors such as aging, ethnicity, genetic predisposition, oxidative stress, and mitochondrial dysfunction also contribute to the development of glaucoma [5,6]. Moreover, in recent years, researchers have also begun to focus on the role of the immune system in glaucoma, particularly neuroinflammation.

Neuroinflammation refers to an inflammatory response that occurs in the central nervous system in response to various insults or diseases. In the context of glaucoma, neuroinflammation involves the activation of resident immune cells and glia in the retina and optic nerve head, such as microglia and astrocytes, as well as the infiltration of immune cells from the bloodstream. Multiple studies have indicated the involvement of various molecular pathways in the pathogenesis of neuroinflammation and the development of glaucoma. These pathways include the tumor necrosis factor α signaling pathway [7,8,9,10,11], the FasL-Fas signaling pathway [12,13], and the Toll-like receptor signaling pathway [14,15,16]. Furthermore, several inflammatory cytokines, interferons, interleukins, and proteins involved in antigen presentation to T cells have also been found to play a role in glaucoma [17,18,19]. However, the complement system has emerged as a key player in glaucoma-related neuroinflammation [20,21,22]. The complement system is a complex cascade of proteins that plays a role in immune defense and inflammation. Dysregulation of the complement system is strongly associated with the degeneration of RGCs, their axons, and synapses in glaucoma [23,24,25,26,27,28]. Moreover, dysregulation of the complement system has also been implicated in many other neurodegenerative diseases, including age-related macular degeneration, diabetic retinopathy, traumatic brain injury, ischemic brain injury, multiple sclerosis, Alzheimer’s disease, and Parkinson’s disease [29,30,31,32,33,34,35,36,37]. These findings highlight the crucial role of complement dysregulation in both local and systemic neurodegenerative processes.

Hence, gaining insights into how complement dysregulation and neuroinflammation contribute to the pathogenesis of glaucoma is important for identifying potential therapeutic strategies. By targeting specific elements of the complement system or modulating neuroinflammatory responses, it may be possible to develop interventions that preserve vision and halt disease progression. Therefore, the objective of this review is to present the current understanding of the complement system’s role in the pathogenesis of glaucoma.

## 2. Overview of the Complement System

The immune system is made up of innate and adaptive systems. The innate immune system provides immediate and non-specific responses to the presence of pathogens, making it the first line of defense against infection. The complement system serves as a crucial component of the innate immune system and plays a role in recognizing pathogens or pathogen-infected cells, tagging them for elimination, and inducing cell lysis, apoptosis, and inflammatory responses [38].

The complement system consists of more than 40 plasma proteins and membrane-bound proteins that interact with each other to opsonize pathogens or damaged host cells and initiate a cascade of inflammatory responses to fight infections. Most of these complement proteins are synthesized as inactive precursors in the liver and are distributed via the bloodstream. But complement proteins are also produced by immune cells, such as tissue macrophages and monocytes, as well as many retinal cells, including microglia, Müller cells, and retinal pigment epithelium (RPE) [39,40].

The complement system consists of three pathways: the classical, lectin, and alternative pathways (Figure 1). An overview of the complement components and their functions is presented in Table 1. The pathways differ in the manner by which they are activated, each triggered by different pattern recognition molecules [41]. However, all pathways lead to the proteolytic cleavage of central complement component C3 (C3), complement component C5 (C5), and the assembly of the membrane attack complex (MAC/C5b-9). Once the complement system is activated, it carries out three main effector functions. Firstly, it generates anaphylatoxins, which recruit inflammatory cells to the site of infection. Secondly, it opsonizes the surface of target cells by binding complement opsonin, such as C3b. Finally, it induces cell lysis through the assembly of the MAC in the cell membrane of the target cell [42].

### 2.1. Classical Pathway

In the classical pathway (CP) of complement activation, complement component 1Q (C1Q) is a pattern recognition molecule that recognizes an array of self, non-self, and altered-self ligands. C1Q is best known for its ability to bind IgG- and IgM-containing immune complexes. In addition, it recognizes molecules such as C-reactive protein (CRP) present on the surface of the pathogens or apoptotic cells [42,43] (Figure 1). The CP is activated when the large protein complex C1, consisting of C1Q, complement component 1R (C1R), and complement component C1S (C1S) serine proteases, recognizes and binds to the fragment crystallizable (Fc) region of the antibody attached to the surface of pathogens or apoptotic cells [44]. This binding triggers a conformational change in the C1 complex, leading to the autocatalytic activation of C1R. Activated C1R cleaves its associated C1S to generate an active serine protease that cleaves complement C4 (C4) to a larger fragment C4b and a smaller fragment C4a [38]. C4b binds covalently onto the target surface, and C4a, an anaphylatoxin, is released into the environment and acts as a potent inflammatory mediator. C1s then cleaves complement C2 (C2) into a large fragment, C2a, and a small fragment, C2b. The C4b, covalently bound to a surface molecule of a pathogen, then binds C2a to form the C4bC2a complex known as the classical pathway C3 convertase, which is responsible for the cleavage of C3 [45]. The cleavage of C3 produces C3a, an anaphylatoxin, and C3b, which acts as an opsonin and will bind to and identify pathogens and apoptotic cells for phagocytosis and removal. Additionally, C3b can also serve to further amplify the complement cascade through binding to complement factor B (CFB) and initiating the alternative pathway, which can act as an amplification loop for all three complement pathways [41].

### 2.2. Lectin Pathway

The lectin pathway (LP) is activated when the pattern recognition molecule mannose-binding lectin (MBL) or ficolin binds to carbohydrates on the surface of a wide range of pathogens [46] (Figure 1). Binding of mannan-binding lectin serine protease 1 (MASP1) to the target surface leads to its autoactivation and subsequent activation of mannan-binding lectin serine protease 2 (MASP2), which then initiates the cleavage of C4 and C2, resulting in the generation of C4a, C4b, C2a, and C2b fragments [47]. Similar to the classical pathway, the C2b and C4a fragments are released into circulation. However, the C4b opsonin can also recruit C2a to form a surface-bound LP C3 convertase, which is identical to the CP C3 convertase C4bC2a.

### 2.3. Alternative Pathway

Activation of the alternative pathway (AP) can be triggered by C3b, generated either by the CP classical convertase or by the “tick-over” through spontaneous hydrolysis of C3 and production of a soluble convertase C3(H_2_O)Bb (Figure 1). When CFB binds to C3b or C3(H_2_O), it undergoes a conformational change. Complement factor D (CFD) then cleaves CFB into soluble Ba and Bb fragments, with Bb remaining bound to C3b or C3(H_2_O) [41]. This complex, known as C3bBb or C3(H_2_O)Bb, forms the AP C3 convertase, which cleaves C3 into C3a and C3b, thus serving as an amplification loop for the other two complement activation pathways [43]. Moreover, while the CP and LP play essential roles in pathogen recognition and initiation of the complement cascade, the alternative amplification accounts for approximately 80% of C5 activation during pathogen recognition [48]. The C3bBb complex is unstable and has a short half-life, but it is stabilized by a protein called properdin (CFP). CFP enhances the amplification properties of the alternative pathway by acting as a stabilizer and directly initiating alternative pathway activation as a recognition molecule [49]. Therefore, the alternative pathway can be considered a recognition pathway facilitated by CFP. The C3bBb complex can further bind an additional C3b molecule, forming the C3bBbC3b complex, which acts as the alternative pathway C5 convertase.

### 2.4. Terminal Pathway

Once the complement system is activated and there is an abundance of C3 convertase, the terminal pathway is initiated. This occurs when there is a sufficient level of C3b that allows C3 convertases from the CP, LP, and AP to bind additional C3b fragments and transform into C5 convertases C4bC2aC3b (for the CP and LP) and C3bBb3b (for the AP) (Figure 1). Therefore, the binding of the C3b to the C3 convertase shifts the substrate specificity from C3 to C5. These C5 convertases then cleave C5, resulting in the generation of C5a and C5b fragments. C5b associates with complement C6 (C6) and complement C7 (C7) to form either a soluble or surface-bound C5b-7 complex. This complex can further bind with complement C8 (C8) and multiple complement C9 (C9) proteins, leading to the formation of either a soluble or surface-bound MAC. Assembly of the MAC leads to the formation of pores that disrupt the cell membrane of the target cells, resulting in cell lysis and death. The MAC, also known as the terminal complement complex, circulates in the bloodstream. MAC binding to the complement inhibitors vitronectin (VTN) or clusterin maintains the soluble form, resulting in the creation of lytically inactive sC5b-9 by preventing the attachment of the complex to the cell surface [50,51]. This binding helps regulate the lytic properties of the MAC.

## 3. Complement System and Inflammation

Under normal physiological conditions, the proteins that make up the complement system largely exist as inactive precursors or zymogens. Cleavage or activation of these precursors is tightly regulated by soluble inhibitors like complement factor H (CFH), C4-binding protein (C4bp), and membrane-bound regulatory proteins such as CD46, CD55, and CD59 [52,53] (Figure 1). These regulatory factors play a crucial role in maintaining the balance of complement activation, preventing excessive or inappropriate complement activity that could lead to tissue damage or immune dysregulation. By inhibiting complement activation at various stages, these regulatory proteins help ensure the proper functioning of the immune response and tissue homeostasis.

The retina is an immune-privileged tissue and possesses unique immune defense mechanisms, including microglia, the resident immune cell, astrocytes and Müller cells, and the complement system [54]. During the natural aging process, there is a phenomenon known as “para-inflammation,” where innate immune cells sense tissue stress and mount an immune response with the goal of restoring tissue homeostasis [55]. This low grade “para-inflammation” can also be observed in the aging retina with increased glial activation, breakdown of the blood-retinal barrier, upregulation of inflammatory genes involved in cytokine/chemokine production, and complement activation [56]. A controlled activation of the complement system is essential for maintaining retinal health and normal function. However, dysregulation or over-activation of the complement system can lead to destructive inflammation and contribute to the development of age-related retinal diseases [57].

The complement system plays a critical role in regulating inflammatory and immunological processes through the generation of complement activation peptides and the MAC. The main complement activation peptides are C3a and C5a, which are polypeptides produced through the cleavage of C3 and C5. C3a and C5a play a regulatory role in neuroinflammation and act as anaphylatoxins that can bind to specific cell surface receptors, namely C3AR1 and C5AR1 (CD88). These receptors can be expressed by the glia, neurons, and infiltrating immune cells in the central nervous system [58]. C3a and C5a can also act as chemoattractants for neutrophils, mast cells, and lymphocytes [59]. In addition, C3a and C5a have been implicated in smooth muscle contraction, increased vascular permeability, and the production of various cytokines such as IL-1β, IL-8/CXCL-8, CCL5, IL-6, and TNFα [60,61]. The C3 cleavage product C3b is gradually converted into iC3b by complement factor I (CFI), which can then bind integrin receptors such as complement receptors CR3 (CD11b/CD18) and CR4 (CD11c/CD18) to trigger cellular phagocytosis, cytokine release, and leukocyte trafficking [29].

The other key component of the complement pathway involved in regulating the inflammatory process is the assembly of the MAC. The MAC is classically known for inducing cell lysis, but in the absence of cell lysis, the MAC can induce the production of pro-inflammatory cytokines such as IL-1, TNFα, CCL2, INF-γ, and IL-8 (CXCL8) [62,63,64]. Moreover, the activation of the MAC at sublytic levels can also enhance inflammatory signaling pathways, including NFκB and inflammasome pathways [65,66]. These findings highlight the intricate involvement of the complement system in regulating inflammation as well as its role in immune regulation and maintaining tissue homeostasis.

## 4. Complement Expression in the Retina

How the complement system is regulated within the retina is still not fully understood. Various studies investigated the expression of complement components within the human retina and mouse retina [40,67,68,69,70]. Recent single-cell RNA sequencing data of mouse retina revealed that each retina cell type expresses a specific signature of complement components [40]. This study showed the main sources of complement in the mouse retina are Müller cells and RPE [40]. Müller cells are the main contributors of *C1s*, *C3*, *C4*, and *Cfb*, while retinal neurons dominate in the expression of the complement regulators *Cfi* and *Cfp*. The RPE expresses mainly the negative alternative complement regulator *Cfh*, but it also expresses *Cfb*, *Cfd,* and the MAC components *C5*, *C6*, *C7*, *C8*, and *C9*. The RPE cells and microglia were found to predominantly express inhibitory complement components (*Cfh* and *Cfi*). On the other hand, neurons and Müller cells were observed to primarily express complement activators, including *C1s*, *C3*, *Cfb*, and *Cfp*. The components of the MAC complex are rarely expressed in the retina, but elevated MAC is observed in human RPE-choroid, associated with aging and age-related macular degeneration [71,72]. It is notable that cell populations with lower abundance in the retina [73], such as microglia and Müller cells, are the major source of complement activators [40].

Taken together, the cell-specific expression of complement components reveals that multiple cell types contribute to the maintenance of complement homeostasis in the retina. Due to the retinal blood barrier, the entry of complement components from the blood into the retina is normally restricted [74]. However, when the retinal blood barrier is broken by injury or disease, allowing an influx of complement proteins, the locally produced complement regulators help to modulate complement activation within the retina.

## 5. Complement in Aging Retina

Aging is a significant risk factor for glaucoma, and studies have demonstrated alterations in complement expression within the retina during the aging process [40,75,76]. Additionally, Mukai et al. showed the complement system is essential for maintaining retinal integrity, and disruptions in complement activation can contribute to the development of inflammatory diseases [75]. Pauly et al. showed that over-activation of the alternative pathway may be a major contributor, with a significant increase in *Cfb* transcripts found in the microglial cells of aged mice [40]. Interestingly, another study also showed an increase in *Cfb* and *C3* expression in the retinal microglia of aging mice, suggesting that aging retinal microglia might be the source of increased complement expression and may contribute to age-related immune dysregulation [77]. However, neurons were identified as the most significant source of *Cfp* transcripts in the retina, and expression increases with age [40]. In contrast, the expression of *Cfh*, a key negative regulator of the complement system, decreases by 50% across all retinal cells during aging. Interestingly, these changes in complement expression did not lead to retinal cell loss but were accompanied by increased microglia numbers and microglia activation [40]. Overall, the aging process enhances the expression of *C1s*, *Cfb*, *Cfp*, and *Cfi*, while *Cfh* expression decreases within the retina [40]. Furthermore, protein expression levels of complement components such as C1q, C3, C4, and Cfb have been found to increase in the retina with age as well [57,77]. A study by Mukai et al. revealed that young and aged complement component-deficient mice (*C1q^−/−^*, *Mbl^−/−^*, *Cfb^−/−^*, *C3^−/−^*, and *C5^−/−^*) exhibit progressive retinal degeneration. Each knockout strain, as compared to young mice, exhibit retinal degeneration characterized by reduced electroretinogram amplitudes and thinning of the inner nuclear layer, indicating diminished retinal function and neuronal cell loss respectively. Interestingly, there were no significant differences observed between young and aged wild-type (C57BL/6) mice [75]. Furthermore, a decrease in inner plexiform/ganglion cell layer thickness, also an indicator of neurodegeneration, was detected in aged *C1q^−/−^*, C*fb^−/−^*, *C3^−/−^*, and *C5^−/−^* mice as compared to the young mice from each knockout strain. No significant difference was observed between young and aged wild-type mice. These findings indicate the complement system plays a crucial role in maintaining normal retinal integrity during the aging process. Dysregulation of the complement system during the aging process will likely disturb tissue homeostasis, resulting in inflammation and potential damage to retinal tissue.

## 6. Complement Dysregulation in Glaucoma Patients

### 6.1. Complement Dysregulation in Glaucomatous Retina

Evidence that the complement system is involved in glaucoma comes from studies demonstrating increased expression of complement proteins in glaucomatous donor eyes as compared to age-matched control eyes without glaucoma. For instance, research by Tezel and colleagues in 2010 revealed that CFH, an inhibitor of the alternative complement pathway, is reduced in patients with glaucoma [20]. In addition, there was an increased expression of complement components (C1S, C1R, C1Q, C3, C4B, C7–9, MASP1, MASP2), and complement receptors (CR1, CR2, C5AR) with a concomitant decreased expression of complement regulators (CFH, C4BP, and clusterin) detected in glaucomatous retinas. With immunolabeling, the most significant increase in complement proteins and receptors was observed in the inner retina, specifically in the RGCs and inner plexiform layers. Additional studies further confirm the increased expression of the classical components C1Q [21,22] and C3 [21] in human glaucomatous retinas. Increased deposition of C1Q, C3, and MAC is observed in the nerve fiber layer and RGC layer of human eyes with glaucoma and even in eyes with ocular hypertension but no evidence of optic nerve damage [21].

Furthermore, an increased expression of C3 has also been detected in human donor retinas with ocular hypertension, along with decreased expression of complement regulators, C1Q-binding protein (C1QBP), and C1-inhibiting factor (C1-INH) [78]. The similarity of this observation to the glaucomatous human retina suggests its relevance to the early synaptic elimination observed in ocular hypertensive retinas [20,21]. It has been suggested that complement activation plays a role in the early loss of RGC synapses in experimental glaucoma [23,24]. Therefore, it would be intriguing to investigate whether complement-mediated tissue clearance is involved in the elimination of dysfunctional RGC synapses in the ocular hypertensive human retina and whether complement-mediated collateral damage to RGCs precedes the development of glaucoma in humans.

### 6.2. Complement Dysregulation in Aqueous Humor and Serum in Glaucoma Patients

Multiple proteomic studies demonstrate significant alterations in complement proteins in the aqueous humor and serum of glaucoma patients, implicating the complement pathway in the pathogenesis of glaucoma [79,80,81,82,83,84,85,86].

#### 6.2.1. Primary Angle-Closure Glaucoma

A recent study examined the proteome profile of the aqueous humor of glaucoma patients as compared to age-matched control cataract patients [84]. The study especially focused on different types of glaucoma including primary acute angle-closure glaucoma (PAACG), primary chronic angle-closure glaucoma (PCACG), and neovascular glaucoma (NVG). The study, among others, revealed that the activation of the immune response is related to glaucoma. They found a significant increase in the expression of complement proteins (C1R, C2, C4A/C4B, C5, C6, C8A, C9, CFB, CFI, and VTN) in glaucoma patients as compared to the control cataract patients.

Interestingly, in older women (≥60 years) with primary angle-closure glaucoma (PACG), a higher risk of progressive visual field loss was found to be significantly associated with decreased serum levels of C3, C4, and C1Q at baseline [87]. However, the same association was not observed in younger women (<60) or men (<60 and ≥60 years). It was proposed that changes in the sex hormone 17-β-estradiol, which is anti-inflammatory [88] and significantly reduced in older women with PACG and progressive visual field loss [89], may explain the differences in the association of C3, C4, and C1q with visual field loss between older women as compared to young women and men. However, while C3, C4, and C1q levels at baseline may serve as biomarkers for predicting visual field loss in older women with PACG, further research is needed to fully elucidate the connection between sex, age, and the complement system.

#### 6.2.2. Primary Open-Angle Glaucoma

Significant dysregulation of complement proteins was observed in the aqueous humor of individuals with progressive POAG as compared to control individuals with cataracts [90]. The complement proteins significantly upregulated included C1QB, CFI, C9, VTN, and complement C8 alpha chain (C8A), while C4b binding protein alpha (C4BPA), CFH, C5, C6, and C7 were downregulated. This shift in the expression of complement activators and regulators in the aqueous humor of progressive POAG patients coincides with a shift in complement activity in the retina and the death of RGCs. However, it remains uncertain whether these changes in aqueous humor composition are a cause or an effect of altered complement activity in the retina. Interestingly, a study conducted by Vashishtha et al. detected a downregulation of C6 as well as the MAC component C8G in the aqueous humor of POAG samples compared to control individuals with cataract [86].

Complement components C3, C1Q, C8 beta chain (C8B), and VSIG (V-set and immunoglobulin domain containing protein 4) have also been found to be upregulated in the aqueous humor of POAG patients when compared to control individuals with cataract [83,86,91]. Interestingly, significant alterations in the expression of several complement components in the aqueous humor have not only been found in POAG, PCACG, PAACG, and NVG, but also in NTG glaucoma, where Lee et al. reported an upregulation of C7 in aqueous humor NTG patients [92]. Similarly, significant upregulation of the complement components VTN, C3, CFH, ficolin-3 (FCN3), and C4A was also found in the serum of POAG patients [93].

By contrast, another study found a significant decrease in complement C3 levels in the serum of patients with POAG, and this decrease was associated with increased severity of the disease [94]. Others suggest the ratio between complement factors C3a and C3 could serve as a marker for complement activation [85]. In this study, the levels of C3a and C3 were measured in the aqueous humor and serum of glaucoma and control (cataract) patients, and a significant increase in the C3a/C3 ratios in the aqueous humor and serum was only found in POAG patients with progressive disease. Moreover, there was a positive correlation between glaucoma progression and the C3a/C3 ratio in both the aqueous humor and serum. In patients with stable POAG, no increase in the C3a/C3 ratio was observed. Together, these results support a strong link between increased complement activation and glaucoma progression. Moreover, glaucoma progression can be linked to both local and systemic changes in complement activation.

## 7. Complement Dysregulation in Animal Models of Glaucoma

In addition to the human studies described above, complement dysregulation is implicated in multiple animal models of glaucoma, providing further evidence linking complement activation and/or dysregulation to glaucoma progression. As detailed below, several animal studies demonstrate complement activation during early-stage disease, prior to any detectable retinal damage [22,23,95,96,97].

### 7.1. C1q

Increased expression of C1q, the initiating protein of the classical pathway, is detected in the retina of several animal models of glaucoma [13,22,95,98,99,100]. In the DBA/2J model of glaucoma, C1q becomes upregulated in RGCs, as well as in optic nerve head (ONH) microglia and/or macrophages, prior to axon degeneration and death of RGCs [23,24,101]. In the retina, the increased expression of C1q is localized to the inner plexiform layer, associated with postsynaptic connections of the RGCs, and precedes RGC dendrite atrophy and death of RGCs. Other studies report similar results with an early upregulation of C1q in multiple animal models of glaucoma including mice [22,98], rats [21,95,96], and monkeys [22]. In development, C1q plays a central role in synaptic pruning, tagging unwanted or redundant RGC synapses for elimination by microglia [102]. Thus, in glaucoma, it is hypothesized that increased C1q in the retina tags RGC synapses for elimination, leading to excessive or uncontrolled synaptic pruning, dendritic atrophy, and axon degeneration. In support of this hypothesis, Howell and colleagues demonstrated that complement C1q subcomponent subunit A (C1qa) deficiency in DBA/2J mice prevented synaptic pruning, protecting the RGC dendritic and synaptic architecture [24]. In addition, an intravitreal injection of a C1 inhibitor in a rat ocular hypertension model also showed protection against dendritic synaptic pruning [26]. Together, these results support the hypothesis that C1q plays an important role in synaptic pruning and dendritic atrophy during the early stages of glaucoma. Therefore, pharmacological inhibition of the classical pathway holds promise for inhibiting harmful synaptic pruning and neurodegeneration in glaucoma.

### 7.2. C3

In the CNS, both C1q and C3 tag neuronal synapses, as well as damaged or apoptotic neurons, for elimination by microglia and/or infiltrating macrophages [23]. Therefore, it is not surprising that, in addition to an upregulation of C1q, an upregulation of C3 in ONH and retina is also detected in multiple animal models of glaucoma [13,21,95,98,103,104,105]. However, knocking out C3 in the DBA/2J mouse model of glaucoma was not neuroprotective, but rather exacerbated the disease process [28]. This study suggested that C3 may play a beneficial role during the early stages of glaucoma, possibly through activation of the epidermal growth factor receptor (EGFR) signaling pathway in ONH astrocytes [28]. In contrast, another study used a more targeted approach, using CR2-Crry to specifically inhibit C3 activation in the retina of DBA/2J mice [25]. In this study, an intravitreal injection of AAV2.CR2-Crry into DBA/2J mice resulted in reduced C3 deposition in RGCs and significant protection of the RGCs and their axons. This more targeted approach, which inhibits C3 activation in the retina, but does not eliminate C3 systemically, suggests that local C3 activation does contribute to the death of RGCs and axon degeneration. Taken together, the inconsistent findings regarding the role of C3, with knockout exacerbating glaucoma in one study and targeted inhibition providing protection in another, emphasize the complicated and context-specific roles of C3 in the development of glaucoma.

### 7.3. CR3 and C3AR1

CR3 is a receptor for C3b, and its interaction aids in the phagocytosis of particles opsonized with C3b. In the DBA/2J mouse model of glaucoma, the genetic subunits of CR3, encoded by *Itgam* and *Itgb2*, are both highly expressed in the monocytes that infiltrate the ONH early in the development of glaucoma [106]. In addition, a significant increase in *C3ar1*, the receptor for the anaphylatoxin C3a, was also detected in the ONH of DBA/2J mice in the early stages of glaucoma, with expression localized to microglia and infiltrating myeloid cells [28]. It was proposed that C3AR1 serves as a major regulator of microglia reactivity and neuroinflammatory function, and *C3ar1* deficiency in DBA/2J mice reduced axon degeneration and death of RGCs at 10.5 months of age [58]. Therefore, C3AR1 is emerging as an important player in glaucoma, and its role in the activation of microglia and neuroinflammation in the retina offers a new potential therapeutic target.

### 7.4. C5

Complement component C5 is a major effector molecule in all three complement pathways, and when activated, C5 is cleaved into two fragments, C5a and C5b, which have very distinct functions [107]. C5a acts as a proinflammatory anaphylatoxin, involved in the recruitment of immune cells and promoting extravasation into tissues where complement has been activated [108,109,110]. Immune cell recruitment and extravasation have also been identified as crucial events in the early stages of glaucoma [111]. C5b, on the other hand, initiates MAC assembly on membranes in the immediate vicinity of activation. Hence, C5 activation can trigger both inflammation and apoptosis.

The DBA/2J mouse line, a commonly used mouse model for hereditary glaucoma, is *C5* deficient, demonstrating that C5 is not required for the development of glaucoma in DBA/2J mice [112,113]. However, the restoration of C5 expression in DBA/2J mice exacerbates the disease. In the presence of C5, DBA/2J mice develop more severe glaucoma at an earlier age, with significant MAC deposition in the RGCs and dystrophic neurites in the optic nerve [114]. This study implicates the downstream components of the complement cascade in the pathogenesis of glaucoma and suggests inhibition of C5 as a potential therapy for glaucoma. In support of this conclusion, two recent studies demonstrate intravitreal administration of a C5 antibody inhibits complement activation and protects the RGCs and optic nerve axons in an autoimmune glaucoma model [115,116]. In both studies, neuroprotection coincided with reduced MAC formation. However, contrasting results were observed regarding microglia activation and immune cell infiltration, with Reinehr et al. showing no significant effect on microglia activation, while Gassel et al. showed a significant reduction in both microglia activation and immune cell infiltration [115,116]. Together, these studies implicate the importance of C5 activation in the pathogenesis of glaucoma and could be a potential new therapeutic target, but additional studies are required to fully elucidate the relative importance of the downstream cleavage fragments C5a and its receptors C5aR1 and C5b and the initiation of glia activation and inflammation versus MAC formation in the pathogenesis of glaucoma.

### 7.5. MAC Assembly

Various studies provide evidence of increased MAC deposition in the retina, especially within the RGC layer and optic nerve, in animal models of glaucoma [21,27,105,114,117]. Increased MAC deposition in the RGC layer, caused by enhanced activation of the complement system, coincides with the apoptosis of RGCs in a rat model of glaucoma [27]. However, Cobra Venom Factor-induced complement depletion reduced MAC deposition in the RGC layer and inhibited RGC apoptosis, as determined by TUNEL staining [27]. Moreover, reduced RGC apoptosis is also correlated with decreased levels of active Caspase-8 (extrinsic apoptosis) and Caspase-9 (intrinsic apoptosis), which can both be induced by MAC formation [118,119] and are known to play an important role in the death of RGCs in glaucoma [120]. On the other hand, in the presence of complement inhibitors, such as CD46, CD55, and CD59, cells can be resistant to MAC-induced lysis, and instead, MAC triggers cell activation, cell proliferation, and the release of proinflammatory cytokines [121]. Relevant to glaucoma, neurons express low levels of complement inhibitors and are highly susceptible to complement-mediated lysis [122], while astrocytes and microglia express higher levels of complement inhibitors and are relatively resistant to complement-mediated lysis [123]. MAC deposits are detected in the RGC layer and nerve fiber layer in both mouse and rat models of glaucoma, and this deposition correlates with apoptosis of RGCs, as well as increased production of pro-inflammatory cytokines and chemokines [21,98]. Moreover, Reinehr et al. demonstrated that increased MAC deposits in the RGC layer co-localized with astrocytes and macrophages/microglia and coincided with increased expression of pro-inflammatory cytokines and chemokines [98]. Therefore, while MAC deposition plays an important role in the pathogenesis of glaucoma, whether it directly mediates RGC apoptosis through the induction of MAC-mediated cell lysis, or indirectly through glia activation and the induction of neurotoxic inflammation requires further investigation.

### 7.6. Involvement of Lectin and Alternative Pathways in Glaucoma

As discussed previously, the complement system can be initialized through three different pathways: classical, lectin, and alternative. While most glaucoma studies implicate the classical pathway, especially C1q, the lectin and alternative pathways have also been implicated in both IOP-dependent and IOP-independent animal models of glaucoma [28,98,104,105,124]. In an experimental autoimmune glaucoma model, qPCR and immunohistochemistry revealed a significant increase in the expression of lectin pathway-associated MASP2 in both the optic nerve and retina [105]. Similar results were also observed in IOP-dependent models of glaucoma, with an increase in lectin pathway-associated MLB and MASP2 in the retina and optic nerve [98,104,105]. Furthermore, increased expression of alternative pathway-associated CFB was also detected in the retina of IOP-dependent and IOP-independent models of glaucoma [28,98,104,124]. Additionally, oxidative stress, another risk factor for glaucoma, downregulated the expression of the alternative pathway component CFH in rat retinal cells [20]. This downregulation of CFH was observed in human glaucoma as well (see Section 6.1), indicating that oxidative stress may contribute to the complement dysregulation observed in glaucoma [20]. Interestingly, the alternative pathway has also been linked to other neurodegenerative diseases such as age-related macular degeneration [30,31], as well as spinal cord and brain injuries [125,126], while the lectin pathway has been linked to brain ischemia [29]. However, the role of these pathways in the pathogenesis of glaucoma remains largely unknown and requires further investigation.

### 7.7. Complement Components as Biomarkers for Glaucoma Progression

A recent study conducted by Fernandez-Vega et al. using the DBA/2J mouse model of glaucoma identified important changes in immune response-related and complement system proteins in the serum that were associated with the development of glaucoma [127]. These protein alterations showed consistent patterns as the disease progressed, suggesting their potential involvement in the pathophysiology of glaucoma. This study identified a panel of five specific proteins, including C4a, CFH, FCN3, apolipoprotein A4 (APOA4), and transthyretin (TTR), that were used to generate machine-learning models, and the study reported a 78% accuracy rate in predicting glaucoma development in DBA/2J mice using this protein panel [127]. Additional work is needed to see if this type of protein panel could be used to predict glaucoma development in other experimental models, with the ultimate goal being to identify systemic biomarkers that could potentially be used for early diagnosis of glaucoma in the clinic.

## 8. Therapy

Multiple studies have used transgenic or pharmacological approaches to inhibit complement components, demonstrating their potential in protecting against RGC loss in animal models of glaucoma [25,101,114]. By inhibiting complement components, these interventions aim to mitigate the neuroinflammatory response and prevent the degeneration of RGCs, which are critical for visual function. The positive outcomes observed in these animal models suggest that targeting complement components could be a promising therapeutic strategy for glaucoma. However, further research is needed to evaluate the efficacy of these treatments in human glaucoma patients.

Clinical trials targeting complement components have been conducted for other diseases such as age-related macular degeneration and glomerulonephritis [128,129]. In the context of glaucoma, so far, the only complement component being targeted in clinical trials is C1q. Inhibiting complement C1q has shown promising results in a recent phase I clinical trial [130]. Here, they investigated the safety and effectiveness of ANX007, a complement C1q-blocking antibody fragment, as a treatment for glaucoma. The phase 1 trial showed that intravitreal injections of ANX007 were well-tolerated by patients, ANX007 was detected in the aqueous humor for an extended period, and ANX007 successfully blocked C1q activity. These promising results lay the groundwork for future studies to assess whether ANX007 can prevent visual field loss in glaucoma patients.

In addition to directly targeting the complement pathway, this review also highlights a significant correlation between complement dysregulation and aging in the retina. The aging retina, naturally more susceptible to stress, seems also to be more prone to complement dysregulation. In this context, a novel strategy has emerged—reversing cellular aging through epigenetic reprogramming of RGCs. This strategy aims to create a more resilient and youthful RGC that can better withstand the stress induced by elevated IOP and recover visual function loss, as demonstrated by recent studies [131,132].

As investigators continue to work with experimental models of glaucoma to identify new targets for the treatment of glaucoma, when targeting the complement system, it is crucial to assess both the positive and negative effects of complement activation in the retina. The therapy should carefully balance the inhibition of complement activation without compromising the homeostatic functions of complement, such as para-inflammation, which occurs with aging as a mechanism to maintain retinal homeostasis [38,75].

## 9. Summary and Conclusions

In summary, multiple studies in human and experimental glaucoma provide evidence that the complement cascade, particularly the classical pathway, plays a significant role in the development and progression of glaucoma. Dysregulated complement activation in glaucomatous eyes indicates a potential mechanism for the induction of neuroinflammation and the death of RGCs. This complement activation is observed in both human glaucoma and multiple experimental animal models, demonstrating its relevance across different species. However, the question of whether complement dysregulation is a cause or effect of glaucoma is a challenging puzzle to solve. On the one hand, multiple animal studies demonstrate a clear association between complement dysregulation and neurodegeneration in glaucoma. However, most experimental models of glaucoma are dependent upon elevated IOP as a trigger for disease. Thus, while complement dysregulation is observed following elevated IOP and is associated with axon degeneration and the death of RGCs, is complement dysregulation alone sufficient to initiate disease? The observation of complement dysregulation in NTG, where IOP remains within the normal range, adds another layer of complexity. The absence of elevated IOP in NTG prompts further exploration into whether the dysregulation of complement alone is sufficient to initiate disease. Furthermore, exploring the dynamics of complement expression specifically in RGCs during disease progression represents an unexplored research area. Investigating the cell-specific changes in complement expression in RGCs could provide valuable insights into the pathobiology of glaucoma, helping to further unravel the molecular causes behind RGC dysfunction in response to elevated IOP.

Further elucidation of how the complement system is activated in glaucoma and the specific consequence of this activation is crucial for the development of effective therapies. Inhibiting the complement system in experimental models of glaucoma using complement inhibitors or targeting specific complement components has shown promising results in preventing axon degeneration and the death of RGCs. Developing targeted therapies that modulate complement activation or its downstream effects could provide new therapeutic approaches for preserving vision and slowing or preventing glaucoma progression. However, further research is necessary to fully understand the implications of these studies and apply them in a clinical setting.

In conclusion, both experimental and clinical evidence strongly supports the involvement of the complement system in the pathogenesis of glaucoma, and ongoing research in this field has the potential to uncover innovative therapeutic approaches to improve outcomes for individuals with glaucoma.

## Figures and Tables

**Figure 1 ijms-25-02307-f001:**
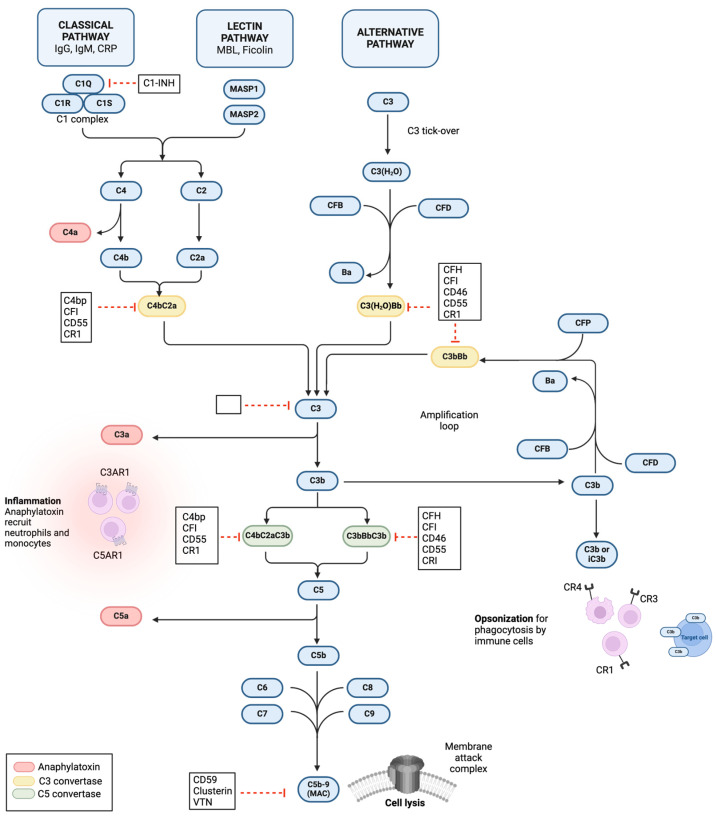
Overview of the complement system and its inhibitors. The complement system is made of the classical, lectin, and alternative pathways. The classical pathway is initiated by the binding of the C1Q complex to IgG, IgM, or CRP present on the target surface, cleaving C4 and C2, and generating the C3 convertase C4bC2a. The lectin pathway becomes activated when MBL or ficolins bind to a carbohydrate on the target surface. The associated MASP1 and MASP2 will then activate the cleavage of C4 and C2 to form the C3 convertase C4bC2a. The alternative pathway can be activated by spontaneous hydrolysis of C3 (tick-over) to form C3(H_2_O), which in the presence of CFB and CFD will lead to the formation of C3 convertase C3(H_2_O)Bb that cleaves C3 into C3a and C3b—similar to the other C3 convertase. C3b is produced by all three pathways and can interact with CFB, CFD, and CFP to form the membrane-bound C3 convertase C3bBb, which serves as an amplification loop for the entire complement system. C3b will be further bound to the C4bC2a or C3bBb complex to form the C5 convertase C4bC2aC3b and C3bBbC3b, respectively. These C5 convertases convert C5 into C5a and C5b, which initiate the terminal pathway. C5b binds with C6, C7, C8, and C9 to form the membrane attack complex (MAC, C5b-9). The MAC forms a pore in the target cell membrane, leading to cell lysis. Cleavage products C4a, C3a, and C5a are anaphylatoxins, which mediate inflammation by recruiting neutrophils and monocytes. In addition, C3b and iC3b act as opsonins, bind to the target cell surface, and mark them for phagocytosis. Figure was created with BioRender.com.

**Table 1 ijms-25-02307-t001:** Components of the complement system and their function.

Components	Name	Main Function
**Classical pathway (CP)**		
IgG	Immunoglobulin G	Binds C1Q, initiates the CP
IgM	Immunoglobulin M	Binds C1Q, initiates the CP
CRP	C-reactive protein	Binds C1Q, initiates the CP
C1Q	Complement C1Q component	Binds IgG, IgM, CRP, initiates the CP
C1R	Complement C1R component	Activates C1S
C1S	Complement C1S component	Cleaves C2 and C4
**Lectin pathway (LP)**		
Ficolin		Binds certain carbohydrates, activates MASP1
MBL	Mannose binding lectin	Binds certain carbohydrates, activates MASP1
MASP1	Mannan-binding lectin serine protease 1	Activates MASP2
MASP2	Mannan-binding lectin serine protease 2	Cleaves C2 and C4
**Alternative Pathway (AP)**		
CFB	Complement factor B	Cleavage fragment Bb is part of the AP C3/C5 convertase
CFD	Complement factor D	Cleaves CFB
CFP	Properdin	Stabilizes the AP C3/C5 convertase
C3(H_2_O)Bb	C3 convertase	Cleaves C3 into C3a and C3b
C3bBb	C3 convertase	Cleaves C3 into C3a and C3b
C3bBbC3b	C3 convertase	Cleaves C3 into C3a and C3b
**Terminal pathway**		
C5	Complement C5	Cleavage fragment C5a is an anaphylatoxin and C5b initiates assembly of MAC
C6	Complement C6	Binds to C5b
C7	Complement C7	Binds to C5b-6
C8	Complement C8	Binds to C5b-7
C9	Complement C9	Binds to C5b-8
C5b-9/MAC	Membrane attack complex	Forms a membrane pore, resulting in lysis of cells, bacteria, and certain viruses
**Classical/Lectin pathway**		
C2	Complement C2	Cleavage fragment C2a is part of the CP and LP C3/C5 convertase
C4	Complement C4	Cleavage fragment C4a is an anaphylatoxin and C4b is part of the CP and LP C3/c5 convertase
C4bC2a	CP and LP C3 convertase	Cleaves C3 into C3a and C3b
C4bC2aC3b	CP and LP C5 convertase	Cleaves C5 into C5a and C5b
**Classical/Lectin/Alternative pathway**		
C3	Complement C3	Cleavage fragment C3a is an anaphylatoxin, and C3b is part of the CP, LP, and AP C3/C5 convertase and opsonin, which can be further cleaved into the iC3b
**Complement receptors**		
C3aR1	Complement C3a receptor 1	Binds C3a
C5aR1	Complement C5a receptor 1	Binds C5a
CR3	Complement receptor 3	Binds iC3b stimulates phagocytosis
CR4	Complement receptor 4	Binds iC3b stimulates phagocytosis
**Membrane-bound regulators**		
CD46	Membrane cofactor protein	Acts as a cofactor for CFI
CD55	Complement decay-accelerating factor (DAF)	Decay-accelerating activity for the C3 and C5 convertases
CD59	CD59 glycoprotein	Inhibits MAC formation
CR1	Complement receptor 1 (CD35)	Decay-accelerating activity for the C3 and C5 convertases, acts as a cofactor for CFI, stimulates phagocytosis
**Soluble-bound regulators**		
CFH	Complement factor H	Decay-accelerating activity to prevent formation of the AP C3/C5 convertase
CFI	Complement factor I	Cleaves C3b and C4b
Clusterin		Prevents membrane insertion of MAC complex
VTN	Vitronectin	Prevents membrane insertion of MAC complex
C1-INH	C1 Inhibitor	Binds to activated C1R and C1S, removing them from C1Q
C4bp	Complement component 4 binding protein	Decay-accelerating activity to prevent formation of CP and LP C3/C5 convertase, acts as a cofactor for CFI

## Data Availability

Data sharing is not applicable. No new data were created or analyzed in this study.

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
