# Peer review of "The Role of Complement Dysregulation in Glaucoma"

_ijms, 2024, doi:10.3390/ijms25042307_

Round 1
Reviewer 1 Report
Comments and Suggestions for Authors
This review explores the role of complement dysregulation in glaucoma pathophysiology. The language throughout the manuscript is concise and easy to understand. Each of the sections are organized and presented efficiently with closing remarks for each of the sections. At times, there are certain sections that could use a bit more specificity in the closing sentences, for example, section 7.1. Nevertheless, I believe that this review could be beneficial for researchers and the readership of the International Journal of Molecular Sciences. Therefore, I recommend the acceptance of this review article with minor revisions.
Major comments:
1. Closing remarks in certain sections seem to present ambiguous and conflicting notions. Additional specificity may help readers organize their comprehension of the concepts presented, especially for those making their way through the article with speed.
2. Figure 1 adequately presents the complex complement system and relevant inhibitors in an organized manner. Considering its substantial complexity, I believe a table presenting the definition/function of key factors discussed in the manuscript could be beneficial to the readership.
3. In section 5, the authors discuss various physiological parameters associated with glaucoma animal models including decreased electroretinogram amplitudes, reduced inner nuclear layer thickness, decrease in inner plexiform/ganglion cell layer thickness. Brief explanations regarding the pathophysiological significance of these changes could present a better picture by connecting them to the increased inflammation as a result of retina aging.
a. Please apply this comment to the clinical data presented at the end of section 6.2.1.
4. The question of whether complement dysregulation in glaucoma is a cause or effect of glaucoma may be a central question regarding their association. Additional discussion regarding this subject may be necessary.
Minor comments:
1. Certain instances where an abbreviation is first introduced are not defined. Please address these.
2. In section 2.1., the authors delve into the role of a number molecules and complexes involved in the classical pathway. I was wondering what the role of C3a, a product of C3 cleavage. Does it play a role in the complement system that could be connected to glaucoma pathology?
3. In page 5 line 159, the authors mention a shift in substrate specificity. Does this refer to the increase in the concentration of C3b? Please clarify.
4. Just out of curiosity has the complement expression of RGCs been researched? This aspect of glaucoma at the cellular level could also be of significant interest in my opinion.
Considering the substantial role the dysregulation of the complement system could pose in glaucoma through its effect on inflammation, this review pinpoints various aspects of the complement system and discusses their specific roles. The concepts presented herein depicts notions of complement dysregulation at the biochemical level to clinical studies. With the potential for a novel class of glaucoma therapeutics, this facet of glaucoma pathology could be of significant interest in the near future. For these reasons, I recommend the acceptance of this review manuscript to the International Journal of Molecular Sciences with minor revisions.
Reviewer 2 Report
Comments and Suggestions for Authors
Glaucoma is a multitude of eye conditions which can impair the optic nerve and lead to vision loss. Complement dysregulation is impaired functioning, an element of the immune system that protects against infections and removes impaired cells. A growing body of literature suggests that complement dysregulation plays a role in the development and progression of glaucoma, especially in a subtype of normal-tension glaucoma (NTG). In this review article, the authors presented it very systematically.
I have a few queries.
- How does oxidative stress play a role in Complement dysregulation?
- How do genetic factors play a role in complement dysregulation?
Reviewer 3 Report
Comments and Suggestions for Authors
The review article with the title “The Role of Complement Dysregulation in Glaucoma” is well-organised and presents a clear overview of the topic. The article discusses molecular pathways especially of the complement system, animal models and therapeutic strategies for the neurodegenerative disease glaucoma and is therefore relevant for International Journal of Molecular Sciences. A geroscience perspective is also included in the article explaining glaucoma and complement dysfunction as an ageing phenomenon. The abstract succinctly summarises the main findings and objectives of the review. The introduction effectively introduces glaucoma as a neurodegenerative disease-condition emphasising pathology and highlights the role of complement in the aetiopathogenesis and development of the condition in the context of neuroinflammation. The amount of background on the topic is well balanced with a comprehensive and clear explanation of the complement system including an illustration of the different pathways in Figure 1. Synthesis and review of the current literature on the topic is broad and includes clinical and basic-science concepts and facts. The conclusion summarises the key points and emphasises the need for further research. The avenues for further research, such as promising directions or conceptual approaches could be further specified. In addition, more detailed discussion and appraisal of advanced therapeutic strategies could be further added to the review article; the article’s therapy section is relatively short. To refer to a geroscience perspective, which conceptually looks at disease as a consequence of ageing of physiological systems, strategies to reverse deterioration and dysfunction of the complement system (anticomplement drugs) could be further discussed. References are relevant for the topic. The editorial statements appear in line with the journal’s guidelines. Overall, the well-written article contributes valuable insights into the role of complement dysregulation in glaucoma, further information, detail and appraisal of the novel therapeutic strategies could be provided.
